# Effects of Motor Imagery Training on Balance and Gait in Older Adults: A Randomized Controlled Pilot Study

**DOI:** 10.3390/ijerph18020650

**Published:** 2021-01-14

**Authors:** Dong Sik Oh, Jong Duk Choi

**Affiliations:** 1Department of Physical Therapy, Division of Health Science, Hanseo University, Seosan 31962, Korea; 2Department of Physical Therapy, College of Health and Medical Science, Daejeon University, Daejeon 34520, Korea; choidew@dju.kr

**Keywords:** motor imagery training, task oriented, balance ability, gait evaluation, fall prevention

## Abstract

The aim of this study was to demonstrate the effects of motor imagery training on balance and gait abilities in older adults and to investigate the possible application of the training as an effective intervention against fall prevention. Subjects (n = 34) aged 65 years and over who had experienced falls were randomly allocated to three groups: (1) motor imagery training group (MITG, n = 11), (2) task-oriented training group (TOTG, n = 11), and (3) control group (CG, n = 12). Each group performed an exercise three times a week for 6 weeks. The dependent variables included Path Length of center of pressure (COP)-based static balance, Berg Balance Scale (BBS) score, Timed Up and Go Test (TUG) score, which assesses a person’s mobility based on changes in both static and dynamic balance, Falls Efficacy Scale (FES) score, which evaluates changes in fear of falls, and gait parameters (velocity, cadence, step length, stride length, and H-H base support) to evaluate gait. After the intervention, Path Length, BBS, TUG, velocity, cadence, step length, and stride length showed significant increases in MITG and TOTG compared to CG (*p* < 0.05). Post hoc test results showed a significantly greater increase in BBS, TUG, and FES in MITG compared with TOTG and CG (*p* < 0.05). Our results suggest that motor imagery training combined with functional training has positive effects on balance, gait, and fall efficacy for fall prevention in the elderly.

## 1. Introduction

As the life expectancy of the elderly increases due to the effects of medical technology and increased living standards, physical changes and health challenges due to aging become a primary concern. Advanced age is associated with decreased flexibility, coordination, and muscle strength, and increased physical response time due to neuronal degeneration. Therefore, the elderly are not capable of rapid response to sudden changes in the environment [1]. These changes in the elderly reduce cognitive motor responses and postural control, thereby increasing the incidence of trauma and death due to falls [2,3]. It has been reported that 30% of the elderly aged at least 65 years residing in the community sustain a fall injury at least once a year. The risks and economic costs of falls increase along with age, and nearly 40% of those who sustain an initial fall injury experience recurrent falls [4,5].

Falls occur due to intrinsic factors related to physiological changes associated with aging or disease as well as extrinsic environmental factors, such as slippery grounds, poor lighting and obstacles [6]. These factors can be independent or related to each other in complex and diverse ways. Decreased ability to maintain posture, increased postural sway, decreased walking speed and dynamic balance may be closely related to the occurrence of falls [7]. The incidence of falls is reported to be the highest during walking, followed by descending stairs, standing up, and changing direction, and is higher among women than men [8].

Recently, various interventions have been attempted to prevent falls in the elderly and improve their daily living activities and ability to balance. Motor imagery training and task-oriented training are being used as evidence-based interventions in diverse fields of rehabilitation [9,10].

Motor imagery training is a learning process in which movements are only internally imagined without being physically carried out [11]. Imagery training engages the same areas of the brain that are activated during exercise, suggesting that the activity of imaginary muscles is increased and resulting in increased muscle strength and speed [12]. For this reason, motor imagery has been used to improve performance in athletes and promote functional recovery in patients at reduced burden and cost, without the need for exercise equipment or therapists [13,14]. In addition, since it can be done at home, it serves as an appropriate intervention in the elderly and patients who cannot participate in daily exercise programs [15]. Multiple forms of motor imagery training exist. However, it is important to include familiar and task-oriented exercise in motor imagery training because the motor neurons are activated further by meaningful exercise-related tasks based on clear-cut goals and motivation [16]. Motor imagery training combined with other therapeutic interventions enhances the recovery of daily living activities in the elderly [17]. Zapparoli et al. reported that the patient’s damaged motor function was recovered by combining motor imagery training and physical therapy [18].

Task-oriented training is based on motor behavior system theory and motor learning. It is an effective intervention in the field of rehabilitation medicine because it provides patients with the motivation to solve problems more actively via exercise programs that are focused on functional tasks rather than repeated and simple training movements. To maximize the effectiveness of task-oriented training, the form of the training task should be similar to that of a task seen in real-world situations, and it should be a meaningful and important task for the subjects. To this end, various forms, such as task-oriented circuit training and progressive resistance task-oriented training, are used as effective interventions in the field of rehabilitation medicine [19,20].

Thus, individual interventions via motor imagery training and task-oriented training are designed to improve balance and gait, respectively. However, most previous studies assessing these interventions involved athletes and patients rather than the elderly, and studies combining the two interventions to assess the risk of falls among the elderly have yet to be reported. Therefore, the purpose of this study was to evaluate the effect of these training on the balance and gait of the elderly after conducting motor imagery training combined with task-oriented training based on various real environments in which falls could occur to prevent falls.

## 2. Materials and Methods

### 2.1. Study Subjects

In this study, 36 healthy elderly persons aged at least 65 years residing at a senior citizen center in an apartment building in S-city, South Korea were assigned to a motor imagery training group (MITG), a task-oriented training group (TOTG), and a control group (CG (n = 12 subjects per group). The final study population consisted of 34 subjects after two dropped out (Table 1). The subjects were randomly assigned to the individual groups using a randomization program (http://www.randomization.com). The calculation was based on a significance level (α) of 0.05 and a power (1 − β) of 0.95 using the G-power 3.1 program, and the effect size was calculated based on the primary effect in previous studies [21].

According to the calculations, a total of 33 subjects were deemed necessary for the study. The selection criteria were: a score of at least 24 points on the Mini-Mental State Examination-Korean (MMSE-K), a history of at least two falls within the one year prior to the study, adequate communication skills, ability to walk independently and to balance, and willingness to provide written informed consent to participate in the study. The exclusion criteria were: brain damage or neurological abnormalities, visual or hearing ailments, treatment with drugs related to walking or balance, participation in programs related to walking or balance at the time of the study, experimental participation rate lower than 80%, or study dropouts (Figure 1).

### 2.2. Study Design and Procedure

The intervention period was 6 weeks long, and training was conducted in 40-min sessions three times per week. During the intervention period, MITG carried out task-oriented training following motor imagery training; TOTG performed only task-oriented training; and CG was educated on fall prevention and health using audio-visual materials. The balance and gait abilities of all subjects were evaluated before and after the intervention, and the Falls Efficacy Scale (FES) scores of the participants were evaluated in relation to these functional performance abilities. Path Length of Center of pressure (COP)-based static balance tests, the Berg Balance Scale (BBS), and timed up and go (TUG) tests were used to evaluate balance. H-H base supports were conducted using the GAITRite system to evaluate gait, velocity, cadence, step length, and stride length. This study was approved by the Institutional Bioethics Committee of Daejeon University (IRB1040647-201910-HR-010-03).

### 2.3. Intervention

#### 2.3.1. Motor Imagery Training

The subjects were asked to relax their bodies and meditate while sitting comfortably in a chair with their eyes closed in a quiet room for 10 min before performing motor imagery training. The subjects then performed motor imagery training for 20 min by freely imagining movements that protected their body and prevented injuries in the event of falls in diverse real-world environments, such as in the bathroom, kitchen, on the stairs, and around obstacles that are frequently associated with falls. During the motor imagery training, a researcher briefly explained the situation verbally using a script that was prepared in advance (Table 2).

#### 2.3.2. Task-Oriented Training

The task-oriented training implemented in this study entailed balance training centered on activities of daily living in the real-life environment of the senior citizen center. In order to reduce the risk of falls and increase the subjects’ interest in the training, the assigned tasks changed over time (Table 3).

### 2.4. Evaluation Tool

#### 2.4.1. Mini-Mental State Examination-Korean (MMSE-K)

The MMSE-K is a tool designed to test cognitive dysfunction in the elderly. The MMSE is relatively easy to use and can be completed in 5 to 10 min. This tool can be used to monitor changes that take place over time because the learning effect of training is small and the tool enables repetitious measurements. The total MMSE score of 30 includes: 10 points for intellectual power, 3 points for memory registration, 3 points for memory recall, 5 points for attention and calculation, 7 points for language skills, and 2 points for understanding and judgment. In this study, additional points were assigned to compensate those who were not educated, though the total score of each individual sub-item never exceeded the full score for that sub-item (e.g., a participant could not receive more than 10 points for intellectual power). A normal (non-cognitively impaired) state is defined by a score of 25 or higher, whereas a score of 21–24 points indicates possible dementia and cognitive impairment is indicated by a score of 20 or lower [22].

#### 2.4.2. Movement Imagery Questionnaire—Revised Second Version: MIQ-RS

The Movement Imagery Questionnaire (MIQ-RS) is designed to assess a subject’s ability to imagine the given tasks. It is composed of a total of 14 items: seven items to evaluate visualization and seven items to evaluate kinematic imagination. Each item is scored on a 7-point scale, ranging from 1 for tasks that are very easy to imagine or implement to 7 for very difficult tasks [23].

#### 2.4.3. Path Length of Center of Pressure (COP)-Based Static Balance Test

Static balance was evaluated using a Wii fit force plate (Balance Board, Nintendo, Kyoto, Japan) and a Balancia software program (Balancia software ver. 2.0, Mintosys, Seoul, Korea). The subjects were asked to stand on the Wii fit force plate with their knees straight and their feet shoulder width apart and maintain a posture with both arms comfortably lowered.

In order to control for postural fluctuations due to eye movements, the subjects were asked to look straight ahead with their eyes open. The measurement was conducted for 30 s beginning with the participants standing on the force plate in stockinged feet with a stable posture. The test was conducted a total of three times and the average value was used. The test was conducted between 2 and 4 p.m. and 10 min of rest were given between each of the three measurements. The center of pressure for the X and Y axes was recorded using the Wii Fit force plate connected to the laptop and Bluetooth technology to measure the forward, backward, leftward, and rightward path length. The Valencia program was used to analyze the data. The sampling rates for data collection were 50 Hz and 12 Hz (using a low-pass filter). The intra-tester reliability of the Wii Fit force plate based on ICC was 0.92–0.98, while the intra-tester reliability of the Valencia program was r = 0.79–0.96 and the validity of the program was r = 0.8–0.96. The Wii Fit force plate and the Valencia program have been validated as useful tools for balance evaluation [24].

#### 2.4.4. Berg Balance Scale (BBS)

BBS is used to evaluate static and dynamic balance. It can be used to predict the risk of falls in elderly subjects without neurological lesions living in the community. BBS is composed of three components: sitting, standing, and postural change. The total score on the BBS is 56 points; it consists of 14 items applicable to daily living activities, each of which is measured with a 5-point scale ranging from 0 to 4 points. A BBS score of 45 or lower indicates that an auxiliary tool such as a cane is necessary when walking, and that the risk of fall is high [25].

#### 2.4.5. Timed Up and Go Test (TUG)

TUG can be used to measure a subject’s ability to balance and move. It entails measuring the time it takes for the subject to walk around an obstacle that is 3 m away from the chair in which he was originally seated and return to the chair as quickly and safely as possible. The test-retest reliability and intra-tester reliability were r = 0.99 [26]. The measurement was performed a total of three times for each participant and the average was used as the measured value. The test was conducted between 9 and 11 a.m. and 10 min of rest were given between each of the three measurements.

#### 2.4.6. Gait Evaluation

To measure temporospatial gait, various gait-related variables including walking velocity, cadence, step length, stride length, double support time, and the base support between the heels of both feet (H-H base support) were measured using the GAITRite system (CIR Systems Inc. Peekskill, NY, USA), which has proven validity. The participants were asked to walk at a comfortable speed 2 m in front of a walking board and on a 4 m walking board according to the examiner’s verbal instructions [27]. The participants were measured indoors with their shoes off and socks on. The gait test was performed three times for each participant and the average value was used. The test was conducted between 1 and 4 p.m. and 10 min of rest were given between each of the three measurements.

#### 2.4.7. Falls Efficacy Scale (FES)

The Falls Efficacy Scale (FES) measures the level of confidence associated with fall prevention. The FES score represents the fear associated with performing 10 actions necessary in daily life on a scale ranging from 1 to 10, and its reliability is r = 0.94. FES score is inversely proportional to the fear of falling and predicts the recurrence of falls. The higher the score, the greater the fear of falls, and the lower the fall efficacy [28].

### 2.5. Analytical Method

The SPSS ver. 21.0 program (IBM, Armonk, NY, USA) was used for all statistical analyses. The general characteristics of the subjects were tested for normality using the Sharpiro-Wilk test. The chi-square test and one-way analysis of variance (ANOVA) were used to test the homogeneity of the general characteristics between the groups and of the pre-experimental results. In order to comprehensively analyze the two factors (treatment application time, intervention method), a two-way ANOVA with repeated measures was performed to assess the effects of interaction between individual factors and each repeated measures factor. Paired t-tests or Wilcoxon signed rank tests were used to analyze changes between measurement time points (before and after intervention) in each group. One-way ANOVA was used to compare the degree of change between groups, and Scheffe tests were used for post-hoc analysis. The significance level was set at α = 0.05.

## 3. Results

### 3.1. Path Length of Center of Pressure (COP)-Based Static Balance Test

Path length of COP decreased significantly more in MITG and TOTG than in CG (*p* < 0.05). Path length was compared between the three groups before and after the intervention; the results indicated significant interactions between the groups and the times (F = 11.076, *p* < 0.05). The changes before and after the intervention in the three groups were significantly different (F = 4.930, *p* < 0.05). According to the results of post-hoc analysis, Path length decreased significantly more in MITG and TOTG than in CG (*p* < 0.05); however, no significant difference existed between the two intervention groups (*p* > 0.05) (Table 4).

### 3.2. Berg Balance Scale (BBS)

Significant increases in BBS score were found in MITG and TOTG after the intervention (*p* < 0.05). The scores of the three groups were compared before and after the intervention, and significant interactions between the groups and the times were observed (F = 22.234, *p* < 0.05). The changes before and after the intervention in the three groups showed significant differences (F = 11.684, *p* < 0.05). According to the results of post-hoc analysis, the scores increased significantly more in MITG compared to the other two groups (*p* < 0.05) (Table 5).

### 3.3. Timed Up and Go Test (TUG)

Significant increases in TUG were found in MITG and TOTG after the intervention (*p* < 0.05). The times of the three groups were compared before and after the intervention, and significant interactions between the groups and the times were observed (F = 12.277, *p* < 0.05). The changes before and after the intervention in the three groups were significantly different (F = 15.607, *p* < 0.05). The results of post-hoc analysis revealed that MITG showed the most significant decrease of all three groups (*p* < 0.05) (Table 6).

### 3.4. Gait Evaluation

Evaluation of gait before and after intervention revealed a significant increase in velocity, cadence, step length, and stride length in MITG and TOTG (*p* < 0.05). The H-H base of support decreased significantly after intervention only in MITG (*p* < 0.05). Analysis of the velocity, cadence, step length, stride length, and H-H base of support in the three groups revealed significant interactions between the groups and the times (F = 12.925, 9.489, 30.784, 23.134, and 16.016, respectively; *p* < 0.05). The changes in velocity, cadence, step length, stride length, and H-H base of support before and after the intervention in the three groups were compared, and the results showed significant differences (F = 8.788, 4.953, 3.596, 4.818, 7.027 *p* < 0.05). The results of post-hoc analysis showed a significant increase in the velocity, cadence, and stride length in MITG and TOTG compared with CG (*p* < 0.05), but no significant difference between MITG and TOTG (*p* > 0.05). According to the results of post-hoc analysis, step length increased significantly more in MITG compared with CG (*p* < 0.05), but no significant difference existed between MITG and TOTG (*p* > 0.05). The results of post-hoc analysis revealed a strongly significant decrease in the H-H base of support in MITG (*p* < 0.05) (Table 7).

### 3.5. Falls Efficacy Scale (FES)

Significant differences in FES score between MITG and TOTG (*p* < 0.05) were found after intervention. The scores of the three groups were compared before and after the intervention, and the results revealed significant interactions between the groups and the times (F = 8.839, *p* < 0.05). The changes between the three groups before and after the intervention were significant (F = 14.017, *p* < 0.05). The results of post-hoc analysis revealed significant decreases in MITG and TOTG compared to CG (*p* < 0.05), and the most significant decrease was found in MITG (*p* < 0.05) (Table 8).

## 4. Discussion

Motor imagery training is consistently reported to improve balance and gait in the elderly, and many studies have investigated falls in the elderly population. Thus, the current study conducted motor imagery training centered on activities of daily life that aimed to maintain balance and protect the body in the event of a fall in order to assess the effects of such training on balance and walking.

Malouin et al. stated that it is important to determine the extent to which subjects can concentrate on imagining movements because the quality of imagination during motor imagery training may differ across individuals [29,30]. This is an important factor that affects the outcome of motor imagery training. Several types of questionnaires that measure the ability to imagine have been developed for a variety of populations. Butler et al. tested whether the MIQ-RS, developed for healthy young adults, was valid for the elderly and stroke patients, and reported that it is useful for evaluating imagination among the elderly and stroke patients [31]. Therefore, in this study, MIQ-RS was used to determine the extent to which subjects imagine or execute movements. The method includes a first-person perspective, in which the person imagines movement by changing the position of the joint angles to coincide with a given activity, and a third-person perspective, in which the person visualizes his/her movements from the viewpoint of others, similar to a movie [32]. Dickstein and Deutsch stated that although the application of these viewpoints varies according to the task at hand, division of the two viewpoints may be academic and artificial [33]. However, Fery argued that the third-person perspective may be effective in imitating forms for writing but applying the first-person viewpoint to the proprioceptive sense is more effective [34]. A study in stroke patients conducted by Kim and Kim reported that the first-person point of view is more effective at enhancing balance than the third-person perspective [35]. Therefore, in this study, since the nature of the task is appropriate for a first-person perspective, the subjects were requested to perform training accordingly.

On the Path Length of COP-based static balance test, the Path Length after intervention decreased more in MITG and TOTG than in the control group, indicating that the ability to balance was improved to a greater extent in the intervention groups. Similar results were obtained on the BBS test, which is designed to evaluate both static and dynamic balances. The TUG, which is another dynamic balance test used to assess rapidity, speed, and agility, also yielded similar results, which are consistent with Hosseini et al.’s findings in stroke patients [36]. Haslinger et al. reported that application of task-oriented visual feedback training in the elderly improved the results of TUG [37]. Liu et al. reported that, in an elderly population, muscle strength and daily activities were improved by a task-oriented exercise program, which coincides with the results of the present study; together, these findings suggest that task-oriented training not only improves balance in the elderly, but also interacts with motor imagery training to result in more positive effects [38]. Whereas MITG and TOTG did not show different results on the static balance test, the group exposed to the combination of motor imagery training and task-oriented training showed more significant improvement than the group that performed only task-oriented training in the dynamic balance test. A study conducted by Shamsipour-Dehkordy et al. reported that combined motor imagery and physical training in the elderly produced more significant improvement in terms of static and dynamic balance than motor imagery training alone [39]. Lee et al. reported that a group of stroke patients exposed to both functional training and motor imagery training showed significantly greater improvement in dynamic balance than a group who performed only functional training [21]. These results are consistent with the results of the present study, indicating that motor imagery is more effective when used together with other interventions. However, according to Boraxbekk et al., neurological response tests in the elderly who learned a new task revealed that the group that combined functional movements with motor imagery showed no greater effect than the group exposed to a single intervention [40]. This phenomenon was explained by activation of mainly the premotor cortex in the movement group and activation of the secondary visual cortex in the motor imagery training group, suggesting that the activities interfered with each other in the combined group, leading to decreased brain activation. Therefore, although motor imagery training has positive effects on balance and gait in the elderly, the results differ depending on the type of task and method used. It may be difficult to activate motor sensation in the elderly when the imaginary movements are identical to actual movements [41], and the ability to learn and maintain new tasks may also decrease in an older population [42]. However, activation of the cerebral visual cortex via imagery training generates muscle tension through voluntary or involuntary movements of the eyes and brain when an object moves or a sudden change in eye direction occurs [43,44]. Also, Mattay et al. stated that, during the elderly learning movement tasks based on visual information, the brain forms more neural networks bilaterally than are formed in younger learners [44]. Therefore, motor imagery training has a positive effect on balance and gait. In addition, since the motor imagery training carried out in this study did not involve new tasks or tasks that included simple movements, but instead focused on tasks that were relatively familiar in daily life and were important in relation to falls, the participants’ motor imaging ability was strengthened to facilitate actual motor functions. In a study of motor imagery training combined with manual dexterity tasks, Ruffino et al. reported that motor ability was improved in the experimental compared to the control group [45]. Based on a systematic review, Nicholson et al. showed that motor imagery training can be an alternative means of improving balance and gait in the elderly who do not have any neurological issues [10].

The walking patterns of the elderly characteristically show reduced velocity, step length, and cadence, wide stride width, long stance phase, and short swing phase [46]. In this study, the GAITRite system was used to evaluate the gait of the elderly participants based on temporospatial variables. The results revealed that both MITG and TOTG showed increased velocity and cadence, which are temporal variables, and step length and stride length, which are spatial variables, compared to the control group. The H-H base support was significantly decreased in MITG. Dunlap et al. reported that the elderly who experience falls show increased width between the two feet when walking in a new environment due to the fear of falling [47]. The results of this study indicated that the H-H base support decreased the most in MITG, resulting in improved walking ability.

Scheffer et al. reported that elderly individuals who experience falls tend to decrease their physical activities due to their fear of falls, which reduces their motor functions, in turn leading to increased risk of falls [48]. Therefore, in this study, Falls Efficacy Scale (FES) tests were conducted to examine changes in the fear of falls in the elderly. The results suggest that the fear of falls decreased the most in MITG, which is consistent with the results of a study conducted by Chung et al. [49]. Kumar et al. reported that fall efficacy is closely related to balance and movement [50]. Therefore, the improvements in balance and walking ability in this study positively affected fall efficacy, and the repetitive learning about falls via motor imagery training further reduced the fear of falls.

Thus, motor imagery and task-oriented training had a positive effect on static and dynamic balance, walking, and fall efficacy in the elderly. Motor imagery training combined with task-oriented training improved functional abilities more than the single intervention. However, there are several problems in interpreting the results of this study. First, because the number of subjects is small, it is difficult to generalize the results to the entire elderly population. Second, no previous studies reported task-oriented training combined with motor imagery training in the elderly. Third, this study failed to evaluate the duration of the learning effect of motor imagery training. Therefore, additional studies are needed to address these limitations.

## 5. Conclusions

This study investigated the effect of motor imagery training on the balance and gait of individuals aged at least 65 years who experienced falls. The results suggest that motor imagery training combined with task-oriented training resulted in significant improvement in static and dynamic balance and walking ability in the elderly, and positively affected fall efficacy. These interventions are simple and cost-effective ways to prevent falls in the elderly. The results of this study represent basic data relevant to the elderly.

## Figures and Tables

**Figure 1 ijerph-18-00650-f001:**
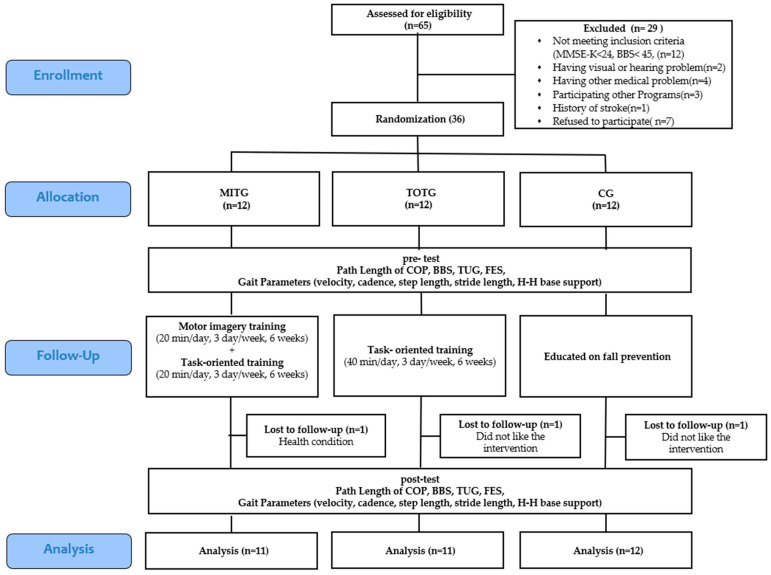
Flow diagram of participant progress. Abbreviations: MITG: motor imagery training group, TOTG: task-oriented training group, CG: control group, MMSE-K: Mini Mental State Examination-Korean Version, COP: center of pressure, BBS: Berg Balance Scale, TUG: Timed Up and Go Test, FES: Falls Efficacy Scale.

**Table 1 ijerph-18-00650-t001:** General characteristics of the subjects (n = 34).

Variables	MITG (n = 11)	TOTG (n = 11)	CG (n = 12)	χ²/F
Gender (Male/Female)	4/7	3/8	4/8	0.095
Age (years)	79.90 ± 5.60 ^b^	78.70 ± 2.62	77.20 ± 2.89	0.404
Height(cm)	157.40 ± 9.66	154.40 ± 6.34	156.10 ± 7.40	0.360
Body Weight (kg)	65.80 ± 9.30	60.80 ± 8.31	61.10 ± 9.70	0.944
MMSE-K(score) ^a^	25.20 ± 1.61	25.30 ± 1.15	25.10 ± 1.28	0.053

^a^ MMSE-K: Mini Mental State Examination-Korean Version, ^b^ Mean ± Standard deviation, MITG: motor imagery training group, TOTG: task-oriented training group, CG: control group.

**Table 2 ijerph-18-00650-t002:** Motor imagery training.

(1) At the moment you get out of bed at night and walk through the dark room, you lose your balance and are about to fall. At this time, reach out to the wall or the floor to protect your body.
(2) At the moment you walk out of the room and sit on the kitchen chair, you are about to fall by mistake. In order not to fall over, reach out and hold the table or wall.
(3) When you walk from the kitchen to the living room, there are many items in the house. At the moment you lift one leg to move your foot in order to avoid an item, you lose your balance and are about to fall. At this time, you open your arms to maintain your balance and reach out not to fall.
(4) Walk from the living room to the kitchen. Raise your heels to take down an item on the kitchen shelf. At the moment you take down the item, you lose your balance and are about to fall. At this time, in order not to fall, you reach out and hold the kitchen wall or sink.
(5) Walk from the living room to the bathroom. At the moment you step into the bathroom, you are about to slip and fall. At this time, in order not to fall, you reach out and hold the sink or wall.
(6) Go out the front door of the house and go down the stairs. At the moment you walk down the stairs, you lose your balance and are about fall. At this time, in order not to fall, you reach out and hold the stair handrail.

**Table 3 ijerph-18-00650-t003:** Task-oriented training.

(1) Moving a cup while shifting the body weight laterally sitting on a Swiss ball
(2) Maintaining balance while standing on an air cushion
(3) Maintaining balance while standing on an air cushion and memorizing the words spoken by the researcher
(4) Walk along a straight line drawn on the floor (5 M distance)
(5) Moving along an S-curve while avoiding obstacles (5 M distance)
(6) Moving water cups
(7) Walking up and down stairs
(8) Putting a cup on the shelf
(9) Replacing bathroom towels
(10) Walking on a cushioned mat

**Table 4 ijerph-18-00650-t004:** Path Length in each group, pre- vs. post-test.

		MITG(n = 11)	TOTG(n = 11)	CG(n = 12)	F	(Time × Group) F
Path Length (cm)	Pre	25.72 ± 2.90 ^a^	26.20 ± 2.13	25.48 ± 1.83		11.076 *
	Post	23.49 ± 2.79	23.34 ± 1.74	25.95 ± 1.47		
	Z	−2.803 *	−2.701 *	−1.172		
	Change	−2.23 ± 0.87 ^†^	−2.86 ± 2.25 ^†^	0.47 ± 1.63	4.930 *	

^a^ Mean ± Standard deviation, MITG: motor imagery training group, TOTG: task-oriented training group, CG: control group, * *p* < 0.05, ^†^ significant difference when compared with the CG (*p* < 0.05).

**Table 5 ijerph-18-00650-t005:** BBS score in each group, pre- vs. post-test.

		MITG(n = 11)	TOTG(n = 11)	CG(n = 12)	F	(Time × Group) F
BBS ^a^ (score)	Pre	46.70 ± 2.40 ^b^	45.80 ± 1.39	46.20 ± 2.14		22.234 *
	Post	49.80 ± 2.57	47.20 ± 1.13	45.10 ± 2.51		
	Z	−2.825 *	−2.724 *	−1.897		
	Change	3.10 ± 1.66 ^†,‡^	1.40 ± 0.84	−1.10 ± 1.59	11.684 *	

^a^ BBS: Berg Balance Scale, ^b^ Mean ± Standard deviation, MITG: motor imagery training group, TOTG: task-oriented training group, CG: control group, * *p* < 0.05, ^†^ significant difference when compared with the CG (*p* < 0.05), ^‡^ significant difference when compared with the TOTG (*p* < 0.05).

**Table 6 ijerph-18-00650-t006:** TUG in each group, pre- vs. post-test.

		MITG(n = 11)	TOTG(n = 11)	CG (n = 12)	F	(Time × Group) F
TUG ^a^ (sec)	Pre	13.58 ± 1.67 ^b^	15.59 ± 2.71	14.84 ± 3.79		12.277 *
	Post	9.22 ± 1.55	12.07 ± 1.12	14.97 ± 3.49		
	Z	−2.803 *	−2.701 *	−0.153		
	Change	4.36 ± 2.15 ^†,‡^	3.51 ± 2.75^†^	0.13 ± 1.31	15.607 *	

^a^ TUG: Timed Up and Go Test, ^b^ Mean ± Standard deviation, MITG: motor imagery training group, TOTG: task-oriented training group, CG: control group, * *p* < 0.05, ^†^ significant difference when compared with the CG (*p* < 0.05), ^‡^ significant difference when compared with the TOTG (*p* < 0.05).

**Table 7 ijerph-18-00650-t007:** Comparison of gait parameters in the three groups: pre- vs. post-test.

	MITG (n = 11)		TOTG (n = 11)	CG (n = 12)	F	(Time × Group) F
Velocity (cm/sec)	Pre	86.84 ± 11.18 ^a^	90.14 ± 8.95	84.80 ± 15.28		12.925 *
	Post	98.13 ± 6.98	100.55 ± 8.08	82.06 ± 15.19		
	Z	−2.805 *	−2.497 *	−1.784		
	Change	11.29 ± 8.03 ^†^	10.41 ± 7.64 ^†^	−2.74 ± 4.51	8.788 *	
Cadence (step/min)	Pre	102.82 ± 3.85	102.33 ± 5.35	103.11 ± 7.67		9.489 *
	Post	112.20 ± 6.50	111.99 ± 4.55	104.35 ± 7.62		
	Z	−2.803 *	−2.701 *	−0.968		
	Change	9.38 ± 4.71 ^†^	9.66 ± 5.72 ^†^	1.24 ± 4.15	4.953 *	
Step	Pre	47.75 ± 4.69	45.07 ± 9.58	46.08 ± 8.01		30.784 *
length	Post	56.38 ± 5.54	50.06 ± 11.02	46.39 ± 7.79		
(cm)	Z	−2.803 *	−2.701 *	−0.866		
	Change	8.63 ± 2.45 ^†^	4.99 ± 3.11	0.30 ± 1.11	3.596 *	
Stride	Pre	95.66 ± 6.22	96.55 ± 6.86	91.43 ± 15.78		23.134 *
length	Post	104.72 ± 3.46	105.24 ± 7.65	92.54 ± 15.83		
(cm)	Z	−2.805 *	−2.803 *	−2.497		
	Change	9.05 ± 4.19 ^†^	8.69 ± 2.74 ^†^	1.11 ± 0.97	4.818 *	
	Pre	9.32 ± 2.14	9.48 ± 1.23	8.58 ± 2.26		16.016 *
H-H base	Post	7.25 ± 1.47	9.27 ± 1.36	8.76 ± 1.94		
(cm)	Z	−2.803 *	−1.376	−0.255		
	Change	−2.06 ± 1.07 ^†,‡^	0.20 ± 0.64	0.18 ± 1.09	7.027 *	

^a^ Mean ± Standard deviation (cm), MITG: motor imagery training group, TOTG: task-oriented training group, CG: control group, * *p* < 0.05, ^†^ significant difference when compared with the CG (*p* < 0.05). ^‡^ significant difference when compared with the TOTG (*p* < 0.05).

**Table 8 ijerph-18-00650-t008:** FES in each group, pre- vs. post-test.

		MITG(n = 11)	TOTG(n = 11)	CG (n = 12)	F	(Time × Group) F
FES ^a^ (score)	Pre	36.70 ± 21.37 ^b^	35.90 ± 10.47	36.40 ± 15.17		8.839 *
	Post	15.70 ± 4.16	26.60 ± 4.92	37.20 ± 14.34		
	Z	−2.805 *	−2.668 *	−1.342		
	Change	−21.00 ± 18.69 ^†,‡^	−9.30 ± 7.04 ^†^	0.80 ± 2.20	14.017 *	

^a^ FES: Falls Efficacy Scale, ^b^ Mean ± Standard deviation (cm), MITG: motor imagery training group, TOTG: task-oriented training group, CG: control group, * *p* < 0.05, ^†^ significant difference when compared with the CG (*p* < 0.05). ^‡^ significant difference when compared with the TOTG (*p* < 0.05).

## Data Availability

The data presented in this article are available on request from the corresponding author.

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
