# Peer review of "Effects of Motor Imagery Training on Balance and Gait in Older Adults: A Randomized Controlled Pilot Study"

_ijerph, 2021, doi:10.3390/ijerph18020650_

Round 1
Reviewer 1 Report
This paper aims to do research on effects of Motor Imagery Training of situation of fall risk on balance and gait in older adults, and evaluates on several methods: motor imagery training , task-oriented training and so on. There are changes that could be made to make the results more accessible and clear to readers, in details:
- The manuscript requires more English polishing. The authors are encouraged to proofread the paper and improve the readability. The keywords chosen by the authors is not suitable. The keywords should represent the content of the full-text topic. (keyword “Imagery”).
- Please provide better introduction and related work section, which better reflects the state of the art of intervention methods and effects, especially the SOTA of the intervention method “motor imagery training” for it is mentioned in the title .
- Illustrations needs to be improved .There is no figure to illustrate the area of brain activation or damage. So the conclusions from the data is not convincing enough to me. I suggest the authors give some figures and to give some further discussions. Furthermore, more validation should be given.
Based on the above reasons, I would recommend rejection.
Author Response
- The manuscript requires more English polishing. The authors are encouraged to proofread the paper and improve the readability. The keywords chosen by the authors is not suitable. The keywords should represent the content of the full-text topic. (keyword “Imagery”).
We agree with the reviewer’s assessment.
We have revised and edited the manuscript to a professional English editor, and we send the manuscript as the attached file.
We think “ Imagery” is the most important keyword that represent the content of the full-text topic.
- Please provide better introduction and related work section, which better reflects the state of the art of intervention methods and effects, especially the SOTA of the intervention method “motor imagery training” for it is mentioned in the title .
We supplemented some of the introduction contents and references.
There are many papers on motor imagery training, but there are few studies that combine motor imagery training with task-oriented training in relation to the fall in the elderly.
To help understand we removed the appendix and presented it as a table in the contents of the motor imagery training.
- Illustrations needs to be improved .There is no figure to illustrate the area of brain activation or damage. So the conclusions from the data is not convincing enough to me. I suggest the authors give some figures and to give some further discussions. Furthermore, more validation should be given.
Older adults and patients may need video information for brain activation. But especially in older people, we don't think brain activation in imaging devices (e.g., FMRI) necessarily improves motor function.
Our study evaluated the balance and gait ability of the elderly. The tools used in the experiment are widely used in the field of rehabilitation because of their proven reliability and validity.
We respectfully ask for your understanding.
Thank you for taking the time to evaluate my paper and we look forward to a good result.
"Please see the attachment."

Reviewer 2 Report
This paper studied the effects of motor imagery training on the balance and gait of the older adults, the experiments were well designed and organized, and the results were clearly presented. There are some minor problems needed to be addressed before it can be published:
1. The English of the paper need to be improved;
2. Survey on existing related works are limited;
3. For the study design, why set the training duration to 40 minutes? If the training duration changes, will it affect the test results? It is suggested that the author should study the effects of training duration on the balance and gait of the elderly
4. There is a Korean word which was not translated into English in Appendix A, Motor imagery training stage1
Author Response
1. The English of the paper need to be improved;
We agree with the reviewer’s assessment.
We have revised and edited the manuscript to a professional English editor, and we send the manuscript as the attached file.
2. Survey on existing related works are limited;
We supplemented some of the introduction contents and references.
There are many papers on motor imagery training, but there are few studies that combine motor imagery training with task-oriented training in relation to the fall in the elderly.
3. For the study design, why set the training duration to 40 minutes? If the training duration changes, will it affect the test results? It is suggested that the author should study the effects of training duration on the balance and gait of the elderly
We agree with the reviewer's suggestion.
The intervention time for the motor imagery training varied from 5 to 40 minutes depending on the researchers. Considering the time required to perform a given task, we set a total of 40 minutes, adding 20 minutes of motor imagery training and 20 minutes of task-oriented training.
Time is important, but above all, we think it is important that subjects could concentrate on the tasks within a given time.
We think that additional studies are need in the future to address duration.
4. There is a Korean word which was not translated into English in Appendix A, Motor imagery training stage1
We removed the appendix and presented it as a table in the contents of the motor imagery training.
"Please see the attachment."

Reviewer 3 Report
I think the work is original and pertinent, but I think it would be necessary to better explain the intervention for its understanding.

Author Response
We agree with the reviewer’s assessment.
- We removed the appendix and presented it as a table in the contents of the motor imagery training.
-
We supplemented some of the newer paper and latest bibliography.
"Please see the attachment."

Reviewer 4 Report
Thank you for the opportunity to read this well written, clear and concise study. It was interesting and a pleasure to read.
The introduction is very complete and allows the reader to know the main topic of the research, informs about the purpose and importance of the work in the clinical field, and also answers the question posed in the scientific context. It includes previous works on the subject in question and makes clear the aspects to be detailed, which constitutes the object of the proposed research.
The “methods” section is one of the most fundamental sections of a scientific article with these characteristics and it is well developed and organized. The instruments used to obtain the data are explained clearly, concisely and are referenced. In addition, they have provided appendices that thoroughly explain the intervention carried out in both intervention groups. However, some aspects should be reviewed, as they could alter the veracity of the study, as follows:
2.1. Study Subjects
- Line 78: “The study included 34 subjects finally after all the dropouts”. What were the reasons for the dropouts? You must specify it in this section.
- Lines 80-82: “The calculation was based on a significance level of α of 0.05 and a power (1-β) of 0.95 in previous studies using the G-power 3.1 program, and the effect size was calculated based on the primary effect in previous studies. As a result, a total of 33 subjects were deemed necessary for the study ”. You must add the reference or references of the previous studies on which you have relied to perform the sample calculation.
- Include in the manuscript the Flowchart showing the assignment of participants to the study and control groups.
2.2. Study Design and Procedure
- Please add the type of study you carried out.
- Was the clinical trial registered?
The authors of this article have meticulously explained the results, providing relevant tables to what was explained in the text. On the other hand, they have provided a clear and complete discussion comparing their results with previous studies and arguing the existing differences. Furthermore, the limitations of this research are very clearly presented.
References
Please check references 3 and 26. Relevant information such as pages is missing.
Author Response
-Line 78: “The study included 34 subjects finally after all the dropouts”. What were the reasons for the dropouts? You must specify it in this section.
We agree with the reviewer’s assessment.
We explained the reason in the flow chart in line 103.
-Lines 80-82: “The calculation was based on a significance level of α of 0.05 and a power (1-β) of 0.95 in previous studies using the G-power 3.1 program, and the effect size was calculated based on the primary effect in previous studies. As a result, a total of 33 subjects were deemed necessary for the study ”. You must add the reference or references of the previous studies on which you have relied to perform the sample calculation.
We added references in line 91.
-Include in the manuscript the Flowchart showing the assignment of participants to the study and control groups.
We included the flow chart in line 103.
-Please add the type of study you carried out.
We added a type of study to the title ( Effects of Motor Imagery Training of Situation of Fall Risk on Balance and Gait in Older Adults; A Randomized Controlled Pilot Study).
-Was the clinical trial registered?
Our study was approved by the Institutional Bioethics Committee.
We showed it in line 119.
References
Please check references 3 and 26. Relevant information such as pages is missing.
We checked references and added pages in line 4 and 31.
"Please see the attachment."

Reviewer 5 Report
Dear corresponding Author,
thanks for submitting your paper. It is very interesting and useful for public health.
In the following lines you can find my comments:
-
- Line 144, please correct "The cener of gravity" with "The center of pressure coordinates"
- In the paragraph 2.4.3 please give more details about the sampling frequency of the instruments and moreover please explain better the subjects' position on the platform. It could be interesting to know the time of the acquisition. I suggest you to check this paper https://www.ncbi.nlm.nih.gov/pmc/articles/PMC4396675/ and cite it on these aspects (position and time of day). Moreover, which kind of parameter of the COP was measured? From the results I suppose the path length and not the sway area. Please can you add this information in the paper?
- It is not clear if during the gait evaluation (paragraph 2.4.6) the subjects walked shod or barefoot. There is a difference in stability and variability between the two conditions and it is better to explain citing reference https://www.mdpi.com/1660-4601/17/12/4569
- Paragraph 3.1 line 191 you wrote "but there was no significant difference in the CG (p > .05)". Then in line 194 you wrote "before 193 and after the intervention in the three groups showed significant differences". It is a contrast. Could you provide a better explanation? Same for paragraph 3.2, 3.3, 3.4, 3.5. I understand your point but it should be written better.
Author Response
1. Line 144, please correct "The cener of gravity" with "The center of pressure coordinates"
We corrected it and reviewer can find it in line 165.
2. In the paragraph 2.4.3 please give more details about the sampling frequency of the instruments and moreover please explain better the subjects' position on the platform. It could be interesting to know the time of the acquisition. I suggest you to check this paper https://www.ncbi.nlm.nih.gov/pmc/articles/PMC4396675/ and cite it on these aspects (position and time of day). Moreover, which kind of parameter of the COP was measured? From the results I suppose the path length and not the sway area. Please can you add this information in the paper?
The sampling rates for data collection were 50 Hz and 12 Hz (using a low-pass filter). Reviewer can find it in line 168.
We measured the path length among several parameters of the COP.
We modified a sway distance to path length because sway distance is not sway area, but path length.
The paper reviewer introduced was very interesting.
We added and explained more details in the paragraph 2.4.3
3. It is not clear if during the gait evaluation (paragraph 2.4.6) the subjects walked shod or barefoot. There is a difference in stability and variability between the two conditions and it is better to explain citing reference https://www.mdpi.com/1660-4601/17/12/4569
The participants were measured indoors with their shoes off and socks on. We added it in line 193.
4. Paragraph 3.1 line 191 you wrote "but there was no significant difference in the CG (p > .05)". Then in line 194 you wrote "before 193 and after the intervention in the three groups showed significant differences". It is a contrast. Could you provide a better explanation? Same for paragraph 3.2, 3.3, 3.4, 3.5. I understand your point but it should be written better
We agree with the reviewer's suggestion.
We corrected all the sentences in Paragraph 3 line 214.
Round 2
Reviewer 1 Report
My previous questions have been satisfactorily addressed and answered. The manuscript has improved significantly due to the revised text, added references and comparisons. Therefore, I agree to a publication after minor revisions in IJERPH.
1.the abstract of revised manuscript is inconformity to the publications in IJERPH.
2.In the keyword section, it is suggested that the words be modified to phrases.
3.The authors should evaluate and discuss clearly their contribution in relation to this paper at the end of introduction.
Author Response
1.the abstract of revised manuscript is inconformity to the publications in IJERPH.
We revised the abstract for the publications in IJERPH.
2. In the keyword section, it is suggested that the words be modified to phrases.
We modified the keyword from word to phrase.
3.The authors should evaluate and discuss clearly their contribution in relation to this paper at the end of introduction.
We discussed our contribution in relation to this paper at the end of introduction.
The appropriate changes made in the revised manuscript were highlighted in red.
Thank you for evaluating our paper.
"Please see the attachment."
